# The Impact of the Seasoning Conditions on Mechanical Properties of Modified and Unmodified Epoxy Adhesive Compounds

**DOI:** 10.3390/polym11050804

**Published:** 2019-05-06

**Authors:** Anna Rudawska

**Affiliations:** Faculty of Mechanical Engineering, Lublin University of Technology, 20-618 Lublin, Poland; a.rudawska@pollub.pl; Tel.: +48-81-538-4232

**Keywords:** epoxy adhesive compounds, physical modification, seasoning, temperature, mechanical properties

## Abstract

The aim of this study was to analyse the impact of the adhesive samples seasoning conditions (temperature and time) on selected mechanical properties of four epoxy adhesive compounds (two unmodified and two modified ones). The samples were made of Epidian 53 epoxy resin mixed with the two different amine curing agents in appropriate stoichiometric proportions. A filler in the form of calcium carbonate (CaCO_3_) powder was used as a modifier. The adhesive compound samples were cured for seven days. Six seasoning variants were used. Four of them were related with the seasoning time at ambient temperature of 24 ± 2 °C for: one month, two months, five months and eight months, respectively. Two other variants were related with seasoning at negative temperature (−10 ± 2 °C) for one month. The last variant (F) also included seasoning at ambient temperature (24 ± 2 °C) for five months right after seasoning in negative temperature. Cured and cylinder-shaped adhesive compound samples were subjected to compressive strength tests (according to the ISO 604 standard). The strength tests were performed using a Zwick/Roell Z150 testing machine. Based on the tests, it was observed that both temperature and time of seasoning influenced the adhesive’s mechanical properties. In the perspective of eight months, these changes were relatively minor for the samples seasoned at ambient temperature. The adhesive samples prepared for the tests were especially sensitive to negative temperature.

## 1. Introduction

Epoxy adhesives are currently one of the most often used adhesives in various industries [1,2,3,4,5,6,7]. Thanks to their high chemical resistance and very good adhesion, epoxy adhesives successfully substitute other bonding agents. They belong to the group of adhesives that may be used to join most of the construction materials available on the market, e.g., metals, polymers, glass and concrete. Thanks to a simple modification of their properties, it is possible to select a perfect adhesive for some specific purposes [1,8,9,10,11,12]. Epoxy resins are often used for laminating [13,14,15,16,17] thanks to minor shrinkage during curing, which is not the case for other epoxies. The minor shrinkage is also an argument for using epoxy adhesives for making hybrid joints, e.g., glass–titanium [9]. When joining materials of different properties, the use of epoxy adhesives allows minimising the stresses that would be created if any other type of adhesive were used [12,18,19,20,21,22,23].

However, as with every type of a polymer material, adhesives are also prone to ageing processes [1,2,24]. These processes have detrimental effect on the adhesive properties. Considering the adhesives susceptibility to ageing, certain attempts are made to select the ageing factors and their impact on the adhesives’ properties changes [6,25,26,27]. Forecasting the changes occurring in adhesives under the influence of one or several factors is very important for the possible use of the adhesive in a specific constructional solution. The aim of the above is to minimise the impact of certain ageing factors [28,29,30,31,32,33,34]. Scientific research presents the effects of ageing and environmental conditions on the adhesion strength of both adhesives and adhesive joints. Knox and Cowling [25] investigated the effect of aging on the adhesion strength of single-part structural epoxy paste adhesive AV119 at the interface. The specimens were subjected to accelerate environmental aging at 30 °C and 100% relative humidity (RH). Knox and Cowling [25] emphasised that the most critical aspect of environmental durability testing is the assessment of the test results and the extrapolation of those results to predicted serviced life. Ameli at al. [17] presented the results of fracture experiments of open-faced double cantilever beam (ODCB) specimens and degraded closed double cantilever beam (CDCB) specimens made from aluminium AA6061-T6 adherends and a commercial DGEBA-based heat-cured rubber-toughened structural epoxy adhesive. The specimens were degraded at 60 °C and 95% RH and 60 °C and 82% RH. Patil et al. [30] considered the same environmental conditions of aging as in [17]. They noticed that, at a given temperature, predictions can be made with reasonable accuracy by extending the knowledge of degradation behaviour from one humidity level to another. The effects of environment condition, surface treatment and materials and geometry were studied by da Silva et al. [28], who subjected adhesive joints were to, e.g., 50 °C and relative humidity of 80% for one, one and four weeks. Popineau and Shanahan [23] measured some mechanical properties of structural epoxy adhesive as a function of ageing time. Hu at al. [22] investigated the effects of long-term temperature exposure on the adhesive joint strength of steel–aluminium alloy sheets prepared by three epoxy adhesives. The results show that long-term temperature exposure causes degradation in joint strength and failure displacement. Da Silva and Adams [26] tested adhesive joints suitable for use from low to high temperatures by the combination of two adhesives (mixed adhesive joint), at −55 °C, 22 °C, 100 °C and 200 °C. They described the issue of using multi-modulus adhesives. Bhowik et al. [31] simulated the service conditions of aerospace applications by exposing adhesive joints of titanium sheets joined by ceramic adhesive to low temperature (−80 °C) and elevated temperature (+500 °C) for 100 h. Ashcroft et al. [27] investigated the effect of environment (−50 °C, 45 °C, and 90 °C with dry and wet conditions) on the fatigue behaviour of bonded composite joints. Grant et al. [32] examined single lap joints and T joints tested at −40 °C, +20 °C and +90 °C. The failure load of lap joints under tension at both +90 °C and −40 °C showed the same decrease in joint strength as the adhesive layer increased as those joints tested at +20 °C. An increase in temperature decreased the adherend yield stress and caused a reduction in the failure envelope. Lapique and Redford [10] conducted an ageing test (40 °C in water vapour) on epoxy adhesive samples and the adhesives properties recorded over a 36-day period. They found that the absorbed water works as a plasticiser leading to a softening of the adhesive. Cavalli et al. [11] exposed different epoxy resins samples and timber joints to various conditions and ageing cycles, e.g., high and low temperature treatments cycles, and treatments at different temperature and RH.

Reducing the susceptibility of polymers to the detrimental ageing processes may be achieved by modifying them both chemically and physically. There are the following modification methods [35,36,37,38,39,40,41]: (i) changing the polymers structure by means of copolymerisation with monomers having the groups that are able to resist the oxidation processes; (ii) hindering depolymerisation by blockage of the last group in the chain; and (iii) introducing additives. A vital direction of research is using different kinds of additives aimed at reducing and slowing down the changes occurring in adhesive joints. These additives may be divided into two groups based on their main purpose [42,43,44,45,46,47,48,49,50,51,52,53,54,55,56]: (i) modifying the physical propertiesl and (ii) increasing the compound’s resistance to ageing and degradation. 

Modifiers that help protect the materials from the ageing factors are: flame retardants, light and heat stabilisers, antioxidants, and fungicides [42,43,57,58,59,60]. Every additive shows different physical and chemical properties. It is impossible to use a single universal modifier. The type of additive is not the only factor that has an impact on the compound properties. Another important factor is the degree of the additive’s dispersion in the material [41]. It is important to mix the additive with the adhesive compound evenly so that it all shows exactly the same properties. The additives that are most often used with adhesive compounds are: fillers, plasticisers, surfactants, and compounds that stabilize the polymers [42,44,50,57,61,62,63,64,65,66]. Moreover, different kinds of fillers have different chemical, mechanical and structural properties. Both their roles and scope of application are much differentiated: from changing the mechanical properties and decreasing cost of production, to changing the colour and look of the adhesive compound. 

Powdered calcium carbonate, which was used for the production of two of the four adhesive compounds studied here, has the most positive impact on the materials’ properties [67,68,69,70,71]. Calcium carbonate (CaCO_3_) is a popular and readily available inorganic filler [54,55,59,60,65] which exhibits good dispersion in different polymer matrix [54,67,70,71] and good properties such as chemically passive, non-toxic, highly pure, odourless and tasteless [49,53,62]. Taking into consideration the compounds’ mechanical properties, its addition improves compressive strength of the cured adhesive. It also has a positive influence on the elastic modulus (Young’s modulus). In some rare cases, it may have a negative effect on the impact strength. As with most fillers, calcium carbonate improves thermal properties, especially thermal stability. It can decrease the cost of production without compromising satisfying properties. It has no significant influence on dielectric and chemical properties. 

## 2. Materials and Methods

### 2.1. Characteristics of Adhesive Compounds’ Modification

Four adhesive compounds were subjected to tests: two modified and two unmodified ones. They were made with epoxy resin based on Bisphenol A (Epidian 53—trade name) mixed with two amine curing agents, namely IDA (based on modified cycloaliphatic polyamine) and Z1 (aliphatic amine, TETA (triethylenetetramine)), in appropriate stoichiometric proportions. Epidian 53 epoxy resin and the applied curing agents are commercial products (manufactured by Organika-Sarzyna, Poland [72]). A filler in the form of the CaCO_3_ powder was used as a modifier. Table 1 presents the analysed samples. 

#### Epoxy Resin

Epoxy resin, trade name Epidian 53 (manufactured by Organika-Sarzyna, Poland [72]), which is a base of the epoxy adhesive compound, is of light-yellow colour and viscous liquid consistency. Epidian 53 epoxy resin is a reaction product of bisphenol-A and epichlorhydrin. It is a styrene-modified epoxy resin. The time of Epidian 53 epoxy resin gelling at ambient temperature, after adding the Z1 curing agent in proportion 1:10, is about 200 min. Initial curing occurs after 6–8 h, whereas full curing takes 7–14 days. Epidian 53 epoxy resin is a liquid styrene-modified epoxy resin. It has a low viscosity (900–1500 m·Pas at 25 °C), average reactivity, and high insulation properties. It is produced by thinning Epidian 5 (the basis epoxy resin) with styrene in a amount ranging from 13 to 15 ns [72]. The density of this epoxy resin is 1.11–1.15 g/cm^3^ at 20 °C and the epoxy number is 0.41. Table 2 shows selected physicochemical properties of Epidian 53 epoxy resin. Some specified information is presented in the BN-73/6376-01 branch standard.

### 2.2. Characteristics of Unmodified and Modified Adhesives

#### 2.2.1. Curing Agents

Two amine curing agents were used in the experiments: Z1 and IDA (trade names, manufactured by Organika-Sarzyna, Poland [72]) (Table 3). These curing agents vary in the amine number. The curing agents network very quickly and, due to short distances between active centres of the cure, dense cross-linking of the resin is obtained. Cured resins with them are characterised by excellent chemical resistance and mechanical strength, but limited flexibility and thermal resistance. Z-1 is an amine curing agent (aliphatic amine, TETA (triethylenetetramine)) and is used for curing low molecular weight epoxy resins and adhesive compounds fabricated on their basis. The IDA curing agent is based on modified cycloaliphatic polyamine. This amine curing agent is moisture-resistant. Curing epoxy resins with the Z1 curing agent causes the increase in elasticity and impacts strength of the adhesive compound. It is recommended to use that curing agent when making joints that are prone to deformation. 

#### 2.2.2. Modifying Agent—Calcium Carbonate

Calcium carbonate (structural formula: CaCO_3_) in a form of white fine-crystalline powder was used as the epoxy adhesive compounds’ modifier [73]. The amount of modifier in the adhesive compound (two weight molecules) was selected based on the literature data [66,67,68,69,70,71] and own tests.

### 2.3. Characteristics of the Cured Adhesive Compounds Samples

Cured, cylinder-shaped samples of the adhesive compounds (Table 1) were subjected to strength tests using a cylindrical polyethylene mould in the following dimensions: d = 13 mm and l = 50 mm. Figure 1 shows the real view of the adhesive compound samples. For each series, 10 samples with the real diameter (d) of 12.19 ± 0.06 mm and length (l) of 37.06 ± 1.55 mm were prepared. 

The effect of microparticle calcium carbonate CaCO_3_ dispersion on the epoxy composition (epoxy resin and curing agent) was studied with SEM images. Figure 2 presents the SEM images of examples of cured modified adhesive compound samples.

The microscopic analysis of cured modified adhesive compounds’ samples was examined with scanning electron microscopy using a MIRA 3 TESCAN microscope. The non-homogeneity of modified epoxy adhesives was evident from the surface of samples. The filler agglomeration was also noticed, which was probably caused by the adhesive compound preparation method and the type of filler. SEM images show the modified adhesive compounds cured for seven days, at the temperature of 21 ± 1 °C and the humidity of 23 ± 1%.

### 2.4. Preparation of the Cured Adhesive Compounds Samples

Preparation of the adhesive compounds for the compressive strength tests was completed in the stages presented in Table 4.

Before being filled with the adhesive compound, the inner walls of the cylinder-shaped moulds were covered with a release agent aimed at processing of polymers and rubber (Polsiform, Polish Silicones, Poland [74]). The agent, after being sprayed, created a release layer on the mould surface. It did not bring any negative effects to the adhesive compound properties. Afterwards, this agent was used to separate the cured adhesive from the mould. 

Selected adhesive compounds were prepared in clean containers by mixing particular ingredients in proportion recommended by the producer. The particular ingredients of the adhesive were measured on a laboratory balance and then were mixed for 3 min until blended together. Mechanical mixing was performed with the use of a paddle mixer that allowed mixing all the adhesives ingredients evenly.

The adhesive compound was then applied to the polymer moulds. The shape of samples was represented by filling the cylindrical moulds with the adhesive compound. To obtain parallel cylindrical sample fronts (necessary to conduct strength tests), the material removal processing was carried out. The fronts were planed on the XH 712 G vertical machining centre produced by Technics Poland (MTP), with the use of a cutter head by Sandvik, with a diameter of Ø50 mm and four cutting edges. The processing was performed with the assumed rotational speed of a spindle at 4000 rpm and with hand feed.

### 2.5. Curing and Seasoning Conditions

The curing process was performed at the temperature of 21 ± 1 °C and the humidity of 23 ± 1% for seven days. Afterwards, the samples were subject to six different seasoning variants. Four of them were related with the seasoning time at positive (ambient) temperature of 24 ± 2 °C for: one month (Variant A), two months (Variant B), five months (Variant C) and eight months (Variant D). Two variants (Variants E and F) were related with seasoning at negative temperature (−10 ± 2 °C) for one month (Variant E). Seasoning at negative temperature was carried out in a temperature chamber (a climatic chamber SH-661 (product of ESPEC, Klimatest, Poland [75]). In the case of Variant F, the samples that were exposed to negative temperature for one month, taken out of the climatic chamber and placed at ambient temperature for another five months. Characteristics of the seasoning variants of the modified and unmodified adhesive compounds are presented in Table 5, Table 6, Table 7, Table 8, Table 9 and Table 10. 

### 2.6. Strength Tests

The adhesive compound samples prepared in accordance with the procedure presented in Table 4, exposed to six seasoning variants (Table 5, Table 6, Table 7, Table 8, Table 9 and Table 10), were subjected to the compressive strength tests, according to the ISO 604 standard [76] and with use of the Zwick/Roell Z150 testing machine. The samples were placed between two parallel cylindrical surfaces. 

The crosshead speed was 5 mm/min. The strength tests were performed on 40 adhesive compositions samples (4 test run × 10 samples, taking into account modified and unmodified epoxy adhesives) in six test runs per each seasoning variant (Variants A–F, Table 5, Table 6, Table 7, Table 8, Table 9 and Table 10). Thus, 240 total epoxy adhesive samples were tested. The basic statistics of the results were considered. The mean and standard deviation were determined, rejecting the extreme values (gross errors) of the obtained results.

## 3. Results and Discussion

### 3.1. Mechanical Properties

The obtained test results were analysed thoroughly. The following strength indicators, measured beforehand, were selected for the analysis: compressive stresses at 1× the per cent of unit shortening σ_1_ (Figure 3), compressive strength σ_M_ (Figure 4), compressive stresses at failure σ_B_ (Figure 5). Compressive stress at 1× the per cent of unit shortening is a strength indicator defined as a proportion of force acting on the sample prepared for the purposes of strength tests to the sample cross-sectional area. This indicator was measured for 1% of the test shapes length reduction against the initial samples length. Compressive strength is defined as the highest compressive stresses that the test sample is able to manage while compressed. Due to specific deformation of the adhesive samples, it was not possible to determine the compressive strength of all adhesive compounds prepared. The last analysed strength indicator was the compressive stress at failure. It defines the compressive stress value at which the adhesive sample fails. The analysis of the aforementioned indicators was aimed at determining the impact of the seasoning time and temperature on the mechanical properties of the modified and unmodified epoxy adhesive compounds samples prepared for the tests. 

Among the adhesive compounds seasoned at ambient temperature (Variants A–D), the highest compression value of 4.41 MPa was observed for the E53/Z1 unmodified adhesive compound seasoned for two months (Variant B) (Figure 4). The lowest stresses of 0.57 MPa were characteristic of the E53/IDA/CaCO_3_ modified adhesive compound that was seasoned at ambient temperature for one month (Variant A). The highest stress at which the adhesive samples from Variant A were deformed at 1% was equal to 1.26 MPa and was observed for both modified and unmodified adhesive compounds with addition of the Z1 curing agent. The difference between the lowest and the highest stress value at 1% shortening unit for Variant A was 55%.

For the adhesive compounds of longer seasoning time (Variant B), the highest stresses occurred for the adhesive with addition of Z1 curing agent without the addition of a filler (unmodified) and they are of 4.41 MPa. The lowest value of compressive stresses at 1% unit shortening among the adhesives seasoned for two months at ambient temperature was observed for the E53/IDA/CaCO_3_ adhesive compound (1.16 MPa). A comparable value of compressive stresses at 1% unit shortening was presented by the adhesives with addition of IDA curing agent without any modifier (E53/IDA). The difference in comparison to the adhesive modified with calcium carbonate was 4%.

The adhesives stored for five months at the temperature of 25 ± 2 °C (Variant C) were characterised by a similar compressive stress value at 1% unit shortening. The difference between the highest and the lowest stress values for the adhesive compounds with the IDA curing agent amounted to less than 9%. In the case of the E53/Z1 adhesive, the compressive stresses that were measured at 1% unit shortening were 1% lower than the adhesive compound E53/Z1/CaCO_3_.

Among the adhesive compounds seasoned at ambient temperature for eight months, the highest compressive stresses at 1% unit shortening were observed for the E53/Z1/CaCO_3_ adhesive compound (2.29 MPa). For the adhesive with addition of the very same curing agent but without a filler, the compressive stresses at 1% unit shortening were 1.57 MPa, giving a difference of 31%. The difference in values of the compressive stresses at 1% unit shortening with addition of the IDA curing agent seasoned in the conditions designed with Variant D was less than 9%. 

The cured E53/Z1 and E53/Z1/CaCO_3_ adhesive compounds seasoned for one month at −10 ± 2 °C were characterised by a similar compressive stress value at 1% unit shortening. The difference in the compressive value at the initial compression phase was less than 1%. In the case of the compounds with addition of the IDA curing agent, higher compressive stresses were observed for the adhesives seasoned at ambient temperature. This difference amounted to 34% for the adhesive without an additive and 60% for the compound with addition of calcium carbonate. 

Among the adhesive compounds seasoned at variable temperature (Variant F), the highest compressive stresses at 1% unit shortening were observed for the E53/Z1/CaCO_3_ modified adhesive compound (3.07 MPa). The difference in comparison to the E53/Z1 unmodified adhesive compound was 52%. In the case of the adhesive compounds with addition of the IDA curing agent, the difference in values of the compressive stresses at 1% unit shortening was 23%. When comparing Variants D and F (identical seasoning time), it was observed that, for the adhesive with the IDA curing agent and without calcium carbonate, the measured compressive stresses at 1% unit shortening were higher by 11% for the adhesive seasoned in the conditions assumed for Variant F. The adhesive stored according to Variant F, with addition of the same curing agent (IDA) and modified with calcium carbonate, showed compressive stresses at 1% unit shortening lower by 21%.

As tone month and ambient temperature, the highest compressive strength (74.48 MPa) was represented by the adhesive with addition of the Z1 curing agent and calcium carbonate (E53/Z1/CaCO_3_). The E53/Z1 unmodified adhesive compound seasoned in exactly the same conditions showed 15.6% lower compressive strength (Figure 4). 

For other variants of adhesive compounds seasoned at ambient temperature but for longer periods of time, the modified adhesives with the Z1 curing agent showed higher strength. However, the impact of a modifier in this case was lower: for Variant B, the E53/Z1/CaCO_3_ modified adhesive was stronger than the E53/Z1 unmodified adhesive by 3.6%; for Variant C, the difference was less than 1%; and, for Variant D, the adhesive with addition of a filler was 1.5% stronger than the adhesive without any additive. 

Among the adhesives seasoned at ambient temperature, the highest compressive strength (80.87 MPa) was represented by the adhesive with addition of the Z1 curing agent and calcium carbonate (E53/Z1/CaCO_3_). It was seasoned for eight months. The lowest compressive strength among the adhesives seasoned at ambient temperature was represented by the adhesive with addition of the IDA curing agent and calcium carbonate (E53/IDA/CaCO_3_) that was seasoned for five months (Variant C). Its compressive strength was 33.25 MPa. In the group of adhesives seasoned for five months at the temperature of 25 ± 2 °C, this adhesive showed 56% more compressive strength than the adhesive with the highest compressive strength (the adhesive with addition of the Z1 curing agent).

Considering an identical period of time (one month) and different temperature conditions in which the adhesive was stored, the adhesive compounds with the Z1 curing agent and without modifiers (calcium carbonate) showed similar compressive strength, where the adhesive seasoned at ambient temperature showed 2.5% more strength. The adhesive containing both the Z1 curing agent and calcium carbonate (the E53/Z1/CaCO_3_ modified adhesive) was more resistant to compression at ambient temperature. Its maximum compressive strength was 74.48 MPa, whereas the same compound seasoned at −10 ± 2 °C had a compressive strength of 45.06 MPa, giving a nearly 40% difference. 

The adhesives from Variant F were seasoned in variable conditions. The adhesive with addition of the IDA curing agent (E53/IDA) stored in accordance with conditions assumed for Variant F showed the compressive strength of 37.10 MPa. The adhesive compound seasoned according to the assumptions of the Variant F with addition of both IDA curing agent and calcium carbonate (E53/IDA/CaCO_3_) showed lower compressive strength (31.74 MPa). The difference between the highest compression strength of the adhesive with addition of the IDA curing agent in a modified form (E53/IDA/CaCO_3_) and the one without a modifier (E53/IDA) was 14%. Compressive strength of both compounds with addition of the Z1 curing agent and seasoned according to Variant F was lower in comparison to the same adhesive compounds that were seasoned for eight months but at ambient temperature. For the E53/Z1 adhesive compound, this difference was 3%, whereas, for the E53/Z1/CaCO_3_ adhesive compound, the discrepancy between the compression strength test results was 7%.

Among the samples seasoned for one month at ambient temperature (Variant A), the highest compressive stresses at failure were represented by the E53/Z1/CaCO_3_ adhesive compound (14.70 MPa) (Figure 5). The E53 adhesive compound that was not modified with calcium carbonate was characterised by a 13.8% lower value of compressive stresses at failure. 

For the E53/Z1 adhesive compound, the highest value of compressive stresses at failure (15.00 MPa) was observed for samples seasoned for a longer period of time (two months). For the very same adhesive compound, the observed increase of the compressive stresses at failure in comparison to the Variant A was 15.5%. 

Both adhesive compounds with addition of the Z1 curing agent that were seasoned for five months at ambient temperature showed similar values of the compressive stresses at failure. For the E53/Z1 adhesive compound, compressive stresses at failure amounted to 14.72 MPa and were 1.3% higher in comparison to the E53/Z1/CaCO_3_ adhesive compound. In the case of Variant C, it was possible to measure compressive stresses at failure for the E53/IDA adhesive compound, and they were 10.35 MPa. 

Among the adhesive compounds seasoned at ambient temperature for eight months, the highest compressive stresses at failure were observed for the E53/Z1 adhesive compound (16.30 MPa). For the same seasoning variant (Variant D), the compressive stresses at failure lower by 4.3% were obtained for the E53/Z1/CaCO_3_ adhesive compound. 

Among the samples seasoned for one month at negative temperature, the highest compressive stresses at failure were represented by the E53/Z1/CaCO_3_ adhesive compound (12.90 MPa). All adhesive compounds seasoned according to the assumptions of Variant E showed lower values of compressive stresses at failure in comparison to the compounds stored in the conditions designated by Variant A. For the E53/Z1 adhesive compound, this difference was almost 5% in comparison to the E53/Z1 adhesive compound, whereas, for the E53/Z1/CaCO_3_ adhesive compound, that discrepancy amounted to 12% in comparison to the E53/Z1/CaCO_3_ adhesive compound.

In the case of Variant F, the compressive stresses at failure were measured for the E53/Z1 adhesive compound as 15.20 MPa.

The use of amine curing agents caused quick cross-link and, due to the short distances between the curing agents active centres, they also caused dense cross-linking of the resin. Resins cured with them were characterised by excellent chemical resistance and mechanical strength, but limited flexibility and thermal resistance. A different amine number of amine curing agents was found to affect the strength parameters. The use of a larger amine number might result in greater brittleness of the resin.

The introduction of calcium carbonate into the epoxy resin contributed to a higher compressive strength as the seasoning time increased. The reason for this might be a reduction in brittleness of the resin. Banea et al. [34] showed that, as the temperature increases, the adhesive tensile strength reduces but the ductility increases. Anes et al. [19] emphasised the negative impact of very low temperatures on the adhesive strength and that micro-cracks formation might result from thermal loads below 0 °C, even for adhesives without ageing. The results presented in Figure 3, Figure 4 and Figure 5 confirm this statement. 

### 3.2. Visual Analysis 

The adhesive compounds samples after the strength tests are presented in Table 11 and Table 12.

When analysing the compressive strength tests results, both shape and look of samples after compressed were considered. They were arranged according to the following scheme (starting from the left): adhesive compound with addition of the IDA curing agent; adhesive compound with addition of the IDA curing agent and calcium carbonate; adhesive compound with addition of the Z1 curing agent; and adhesive compound with the Z1 curing agent and modified with calcium carbonate. 

Variant A

Minor scratches with no distinct signs of deformation were observed on the samples of the E53/IDA adhesive compound. The E53/IDA/CaCO_3_ adhesive compounds seasoned according to the assumptions of the Variant A did not show any shape deformation. There were only some minor scratches on the samples’ circumferences. In the case of the E53/Z1 adhesive compound, no scratches or cracks were visible. There were some signs of shape deformation, however, especially buckling. The samples of the E53/Z1/CaCO_3_ adhesive compound after strength tests showed tendency to crack at the circumference from the test shape’s front. 

Variant B

Samples of E53/IDA adhesive compound after the strength tests were characterised by more cracks than the samples of the same adhesive compound seasoned according to the Variant A conditions. There was a whole network of cracks visible on the samples of the E53/IDA/CaCO_3_ adhesive compound. The cracks were much more visible than in the case of the E53/IDA/CaCO_3_ adhesive compound. Samples of the E53/Z1 adhesive compound were more prone to deformation, as revealed by the test shape buckling during compression. The test shape cracked, starting from its face surface. Some cracks were visible on the samples of the E53/Z1/CaCO_3_ adhesive compound. Plastic buckling occurred on the test shape. There were also deep broken-out sections on the sample’s circumference.

Variant C

The sample of the E53/IDA adhesive compound showed visible deformation after compressed. Many deep cracks were visible on the cylindrical surface. On the E53/IDA/CaCO_3_ adhesive compound sample, there was a network of minor cracks. The shape buckling was relatively minor. In the case of the E53/Z1 adhesive compound, the samples’ shape after compression was similar to the initial shape and buckling was hardly visible. No cracks or scratches were visible. Compression caused deep cracks starting from the sample’s face on the cylindrical surface of the E53/Z1/CaCO_3_ adhesive compound. A minor shape deformation occurred here. 

Variant D

Cylindrical surface of the E53/IDA adhesive compound was covered with numerous yet small cracks. Test shapes of the E53/IDA/CaCO_3_ adhesive compound did not show any signs of shape deformation. Cracks and scratches at the circumference were hardly visible to the naked eye. In the two other adhesive compounds (with addition of the Z1 curing agent), deformations were much more visible. Samples of the E53/Z1 adhesive compound showed a tendency to crack at the face, which led to spalling during further compression. Test shapes of the E53/Z1/CaCO_3_ adhesive compound were also characterised by the tendency to crack and spall. However, it was much more intense than in the case of the E53/Z1 adhesive compound.

Variant E

There were no cracks or scratches on the cylindrical surfaces of the E53/IDA adhesive compound samples seasoned for one month at the temperature of −10 ± 2 °C. These samples, after compression on the testing machine, did not show any deformation signs. An identical situation was observed for the E53/IDA/CaCO_3_ adhesive compound. There were no cracks and the test shape buckling was minor. Samples of the E53/Z1 adhesive compound seasoned at negative temperature for one month showed greater tendency to buckling than the adhesive compounds seasoned in identical conditions but with addition of the IDA curing agent. Minor cracks were visible. A significant shape deformation was observed for the E53/Z1/CaCO_3_ adhesive compound. There was also a network of cracks and minor scratches at the shape circumference.

Variant F

The samples of the E53/IDA unmodified adhesive compound showed a tendency to shape deformation. Buckling and scratches at the circumference were also visible. The samples of the E53/IDA/CaCO_3_ adhesive compound were also prone to deformation. Buckling of these samples was to lesser extent than in the case of samples of the E53/IDA adhesive compound. There was also a network of cracks that occurred during compression. Significantly deformed samples of the E53/Z1 adhesive compound showed a tendency to crack and spall during the test. The cracks started from the test shape face. Numerous deep cracks were also visible on the samples of the E53/Z1/CaCO_3_ adhesive compound. In the case of this adhesive compound, the spalling was minor. A shape deformation that resulted in buckling was visible.

The effect of particle size distribution on the brittle ductile transition of high-density polyethylene (HDPE)/CaCO_3_ composites was studied by Liu et al. [62] who claimed that a narrow particle size distribution is favourable to the enhancement of the toughness of polymer composites. Similar results are presented in Table 11 and Table 12. Ghalia et al. [48] noticed that PP/LLDPE blends at temperature of 23 °C show a ductile fracture mode characterised by the co-existence of a shear yielding process, whereas, at lower temperature (−20 °C) the fractured surfaces of specimens appears completely brittle. Comparing Variant A (Table 11) and Variant E (Table 12), where the time of seasoning was the same, it was found that the negative temperature affected the brittleness of the epoxy compounds, where the effect was visible for the modified compounds.

## 4. Conclusions

The aim of the tests was to analyse the impact of the seasoning temperature of adhesive samples on selected mechanical properties of four epoxy adhesive compounds (unmodified and modified ones). Adhesive compounds samples prepared for the test purposes were seasoned at different temperatures for different lengths of time.

After analysing the obtained results, the following conclusions and remarks were drawn:
Compressive strength of the samples seasoned at ambient temperature increased as the seasoning time increased.Adhesives with addition of the Z1 curing agent and calcium carbonate (modified adhesives) showed a greater compressive strength at ambient temperature than the adhesive compounds cured with the Z1 curing agent but without a modifier.The adhesive compound with addition of the IDA curing agent, seasoned for five months at ambient temperature, was characterised by significantly reduced the compressive strength than other adhesive compounds with addition of the Z1 curing agent.In the perspective of seasoning for one month, the seasoning temperature of the non-modified adhesive compounds containing IDA curing agent did not have a significant impact on the compressive strength.Addition of calcium carbonate to the adhesive compound with Z1 curing agent (seasoned for one month at the temperature of −10 ± 2 °C) caused an almost halving of its compressive strength in comparison to the E53/Z1 adhesive compound seasoned for the same period of time but at ambient temperature.Variable seasoning conditions (Variant F) had an insignificant impact on reduction of the adhesives’ compressive strength; this impact was more significant in the case of the E53/Z1/CaCO_3_ modified adhesive compound.Among the adhesive compounds, those with addition of the Z1 curing agent showed a tendency to crack and spall. The longer the seasoning time was, the more intense these changes were.The samples of the adhesive with addition of the IDA curing agent were susceptible to buckling.The negative temperature affected the brittleness of the epoxy compounds, where the effect was visible for the modified adhesive compounds.

To conclude, both temperature and time of seasoning influenced the adhesives’ mechanical properties. In the perspective of eight months, these changes were relatively minor for the samples seasoned at ambient temperature. The adhesive samples prepared for the tests were especially sensitive to negative temperature.

Adhesive joints are often exposed to difficult conditions. When selecting the adhesive, it is of great importance to be aware of the impact of working temperature on the strength properties of the adhesive joints. It is also important to know what changes occur under the influence of various exploitation conditions (e.g., temperature, humidity, etc.) in the context of the amount of time during which the adhesive joint is exposed to specific environmental conditions. 

## Figures and Tables

**Figure 1 polymers-11-00804-f001:**
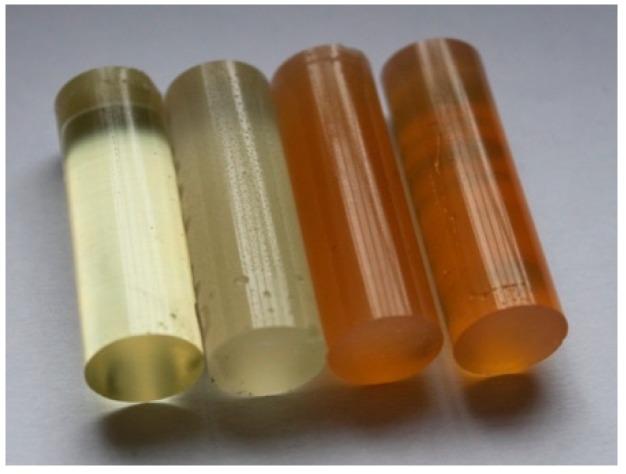
Adhesive compounds’ samples after the curing process.

**Figure 2 polymers-11-00804-f002:**
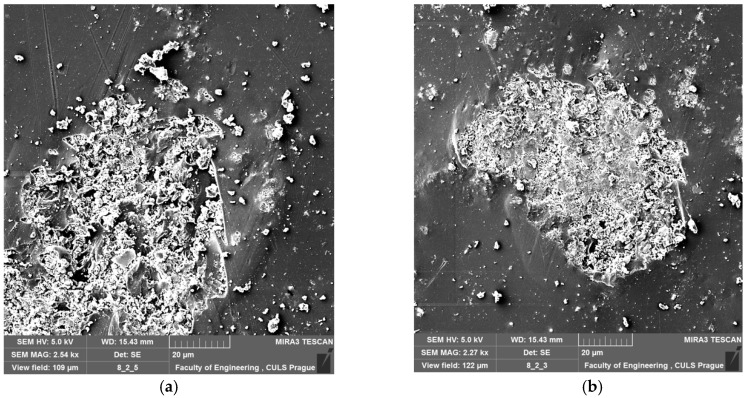
SEM images of cured modified adhesive compound samples: (**a**) E53/Z1/CaCO_3_; and (**b**) E53/IDA/CaCO_3_.

**Figure 3 polymers-11-00804-f003:**
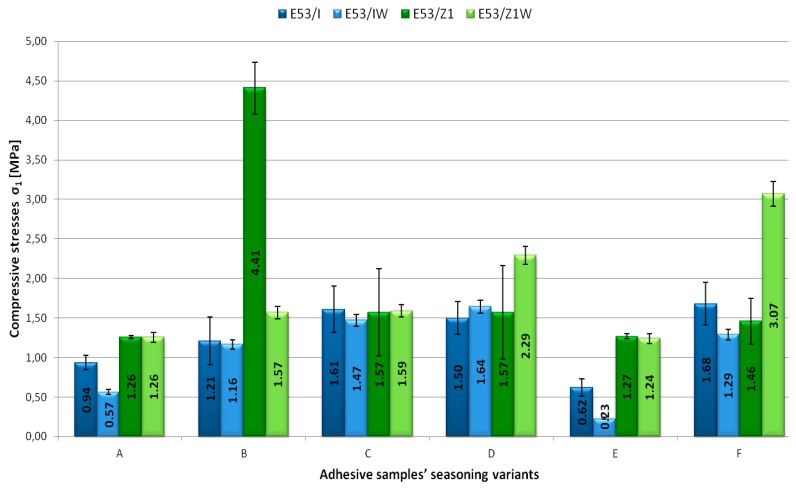
Dependence of the compressive stresses at 1× the per cent of unit shortening σ_1_ [MPa] on the adhesive samples’ seasoning conditions.

**Figure 4 polymers-11-00804-f004:**
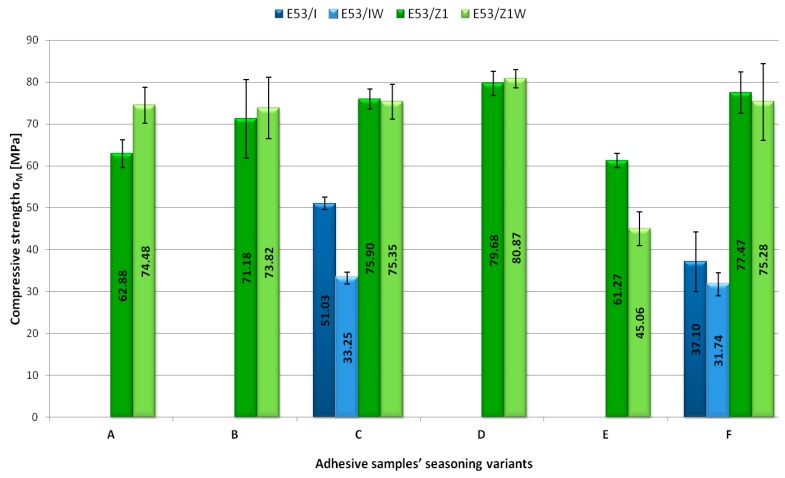
Compressive strength values σ_M_ [MPa] depending on the adhesive samples’ seasoning conditions.

**Figure 5 polymers-11-00804-f005:**
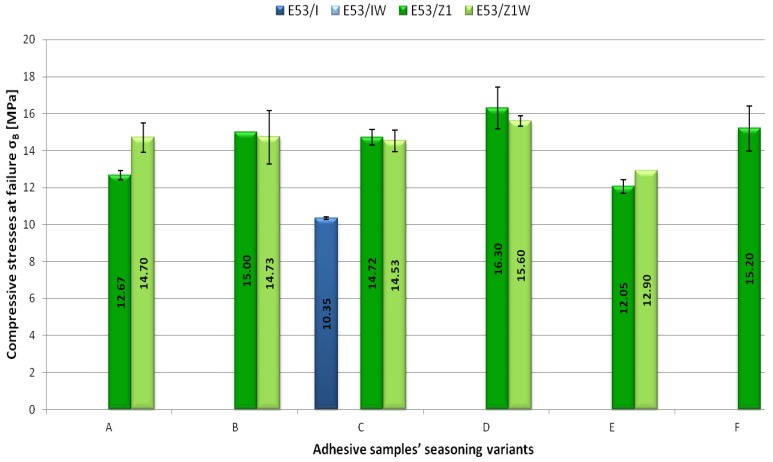
Dependence of the compressive stresses at failure σ_B_ [MPa] on the adhesive samples’ seasoning conditions.

**Table 1 polymers-11-00804-t001:** Types of the epoxy adhesive compounds used in tests.

No.	Adhesive Compound Designation	Resin Type	Curing Agent Type	Curing Agent Weight Per 100 Resin Weight Molecules	Calcium Carbonate Weight Content Per 100 Resin Weight Molecules
1	E53/IDA	Epidian 53	IDA	40	-
2	E53/IDA/CaCO_3_				2
3	E53/Z1		Z1	10	-
4	E53/Z1/CaCO_3_				2

**Table 2 polymers-11-00804-t002:** Physicochemical properties of Epidian 53 epoxy resin [72].

Physicochemical Properties	
State of aggregation	liquid
Color	light-yellow
Smell	characteristic of styrene
Epoxide number [mol/100 g]	0.410
Density at temperature of 20 °C [g/cm^3^]	1.110–1.150
Adhesiveness at temperature of 20 °C [mPas]	900–1500
Time of gelling of 10 g of Epidian 53 + 1.05 g of curing agent Z1 at 20 °C [min]	200
Initial curing [h]	6–8
Full curing [days]	7–14

**Table 3 polymers-11-00804-t003:** Physical and chemical properties of the Z1 and IDA curing agents [72].

Properties	Z1 Curing Agent	IDA Curing Agent
State of aggregation	liquid	liquid
Colour	light-yellow	transparent
Adhesiveness at temperature of 20 °C [mPas]	-	150–300
Density at temperature of 20 °C [g/cm^3^]	0.978–0.983	1.01–1.03
Amine number	Min. 1100 mg KOH/g	200–350 mg KOH/g

**Table 4 polymers-11-00804-t004:** Stages of preparation of the adhesive compounds’ samples.

No.	Stage	Description
1	Forms’ preparation	covering the forms with a release agent waiting for 5 min to let the release agent evaporate
2	Adhesive compounds preparation	measuring the adhesive compound’s ingredients mixing the adhesive compounds ingredients mechanically using a paddle mixer in a polymer container - mixing time: 3 min - rotational speed: 360 rpm - special stand for the adhesives mixing
3	Filling the forms with adhesive compound	applying the adhesive compound into the forms
4	Adhesive compounds curing	curing time: 7 days curing temperature: 21 ± 1 °C humidity: 23 ± 1 °C
5	Seasoning of the adhesive compound samples	seasoning of the adhesive compound samples for the assumed time, at the temperature adequate to the particular seasoning variant (Table 5, Table 6, Table 7, Table 8, Table 9 and Table 10)
6	Material removal processing	a) samples seasoned at ambient temperature: planning of the front of the samples seasoned at ambient temperature after the assumed time b) samples seasoned at negative temperature: taking the samples out of the temperature chamber after the assumed seasoning time 24-h seasoning at ambient temperature: planning of the samples front c) samples seasoned at changing temperature: taking the samples out of the temperature chamber after the assumed time of seasoning at negative temperature seasoning the samples at ambient temperature (for the appropriate amount of time) planning of the samples front.

**Table 5 polymers-11-00804-t005:** Variant A.

No.	Adhesive Compound	Curing Conditions	Seasoning Conditions
1	E53/I/P/1	21 ± 1 °C 23 ± 1%	1 month 23 ± 2 °C 23 ± 1%
2	E53/IW/P/1		
3	E53/Z1/P/1		
4	E53/Z1W/P/1		

**Table 6 polymers-11-00804-t006:** Variant B.

No.	Adhesive Compound	Curing Conditions	Seasoning Conditions
1	E53/I/P/2	21 ± 1 °C 23 ± 1%	2 months 23 ± 2 °C 23 ± 1%
2	E53/IW/P/2		
3	E53/Z1/P/2		
4	E53/Z1W/P/2		

**Table 7 polymers-11-00804-t007:** Variant C.

No.	Adhesive Compound	Curing Conditions	Seasoning Conditions
1	E53/I/P/5	21 ± 1 °C 23 ± 1%	5 months 23 ± 2 °C 23 ± 1%
2	E53/IW/P/5		
3	E53/Z1/P/5		
4	E53/Z1W/P/5		

**Table 8 polymers-11-00804-t008:** Variant D.

No.	Adhesive Compound	Curing Conditions	Seasoning Conditions
1	E53/I/P/8	21 ± 1 °C 23 ± 1%	8 months 23 ± 2 °C 23 ± 1%
2	E53/IW/P/8		
3	E53/Z1/P/8		
4	E53/Z1W/P/8		

**Table 9 polymers-11-00804-t009:** Variant E.

No.	Adhesive Compound	Curing Conditions	Seasoning Conditions
1	E53/I/-10/1	21 ± 1 °C 23 ± 1%	1 month −10 ± 2 °C
2	E53/IW/-10/1		
3	E53/Z1/-10/1		
4	E53/Z1W/-10/1		

**Table 10 polymers-11-00804-t010:** Variant F.

No.	Adhesive Compound	Curing Conditions	Seasoning Conditions
1	E53/I/PM/8	21 ± 1 °C 23 ± 1%	3 months: −10 ± 2 °C 5 months: 23 ± 2 °C 23 ± 1%
2	E53/IW/PM/8		
3	E53/Z1/PM/8		
4	E53/Z1W/PM/8		

**Table 11 polymers-11-00804-t011:** Visual analysis of the shape of the samples seasoned at ambient temperature.

No.	Variant	The View of Samples After Strength Tests
1	A	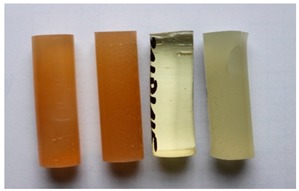
2	B	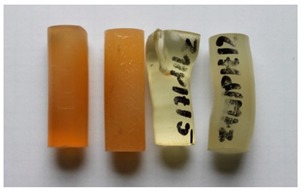
3	C	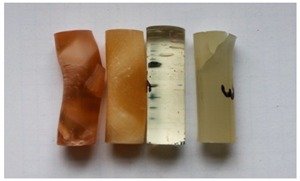
4	D	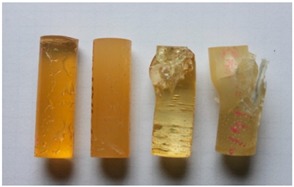

**Table 12 polymers-11-00804-t012:** Visual analysis of the shape of the samples seasoned at negative and changeable temperature.

No.	Variant	The View of Samples After Strength Tests
5	E	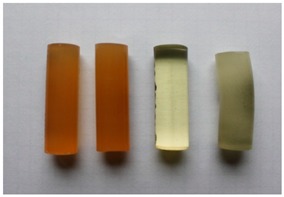
6	F	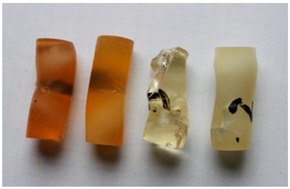

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
