# Peer review of "The Impact of the Seasoning Conditions on Mechanical Properties of Modified and Unmodified Epoxy Adhesive Compounds"

_polymers, 2019, doi:10.3390/polym11050804_

Round 1
Reviewer 1 Report
This manuscript presented a study about the effect of the seasoning conditions on the mechanical properties of epoxy adhesive compounds. The main idea of this work is interesting. However, the author must deeply improve the discussions related to the obtained results in view of the better understand the effects of temperature and time on the structure of the epoxy adhesive compounds, as described below. I suggest major revision.
Lines 127-131: please remove these sentences from the manuscript.
Lines 133-134: remove this sentence from the manuscript.
Lines 137-138: remove this sentence from the manuscript.
Lines 151-154: remove these sentences from the manuscript.
Section 2.1: please better describe the chemical constitution of the curing agents used (IDA and Z1). This information is necessary for better understand the behaviour of the when there are exposure to time and temperature.
Figure 1: I suggest remove this figure from the manuscript.
Figure 2: this figure and the discussion about this figure are results and must be added to the results and discussion section.
Figure 3: I suggest remove this figure from the manuscript.
Section 2.5: how many specimens were used to obtain the mean and standard deviation?
Section 3.1: in this section the author only discussed what samples presented the better behaviour without discuss which caused this behaviour. A deeply discussion about the effect of temperature and time on the chemical structure of the samples is necessary. In addition, the author must compare the results of this work with others from the literature.
Figure 4, Figure 5 and Figure 6: the author must discuss the effect of using CaCO3 and the effect of the different curing agents on the properties evaluated.
Section 3.2: the discussions in this section must be improved. What the usage of different curing agents caused in the properties evaluated? What the effect of the different chemical structure of the curing agents on the properties? What is the effect of using CaCO3? Moreover, the effect of temperature and time on the ageing of the samples?
Conclusions section: please improve this section based on the above comments. In addition, remove lines 489 to 492.
Author Response
Response – Reviewer 1
Thank you for all comments.
I try to consider all Reviewer comments.
Some information was change and deleted.
All changes are in blue.
I would like to answer to the following questions.
Lines 127-131: please remove these sentences from the manuscript.
Lines 133-134: remove this sentence from the manuscript.
Lines 137-138: remove this sentence from the manuscript.
Lines 151-154: remove these sentences from the manuscript.
Response: The sentences were removed from the manuscript.
Section 2.1: please better describe the chemical constitution of the curing agents used (IDA and Z1). This information is necessary for better understand the behaviour of the when there are exposure to time and temperature.
Response: Some information of characteristics of curing agent was completed.
Figure 1: I suggest remove this figure from the manuscript.
Response: Fig. 1a was remove from the manuscript, but the Fig. 1b did not removed from the text, because this figure presents the real view of the adhesive compounds’ samples. I hope that the Reviewer will not mind.
Figure 2: this figure and the discussion about this figure are results and must be added to the results and discussion section.
Response:
The presented SEM images were intended only for the general characterization of the adhesive compositions obtained. It is noted that the non-homogeneity of the resin is evident due to filler agglomeration. Further analysis of the method of preparing the adhesive composition, including the mixing method, will be presented in further publications in which attention will be focused on the methodology of preparing adhesive compositions modified with calcium carbonate. For this reason, this point has been included in the research methodology, as a characteristic of adhesive compositions modified with calcium carbonate, and not as test results. Due to the fact that this is not the subject of the analysis in this article, I decided that these messages will remain in item 2, as an additional description of the analyzed compositions, allowing to display this composition in the cured state.
Figure 3: I suggest remove this figure from the manuscript.
Response: Fig. 3 was removed from the manuscript, and additionally the figures were renumbering.
Section 2.5: how many specimens were used to obtain the mean and standard deviation?
Response: The information of amount of specimens was presented in line 162 and lines 236-239 (“The strength tests were performed for 40 adhesive compositions samples (4 test run x10 samples taking account of modified and unmodified epoxy adhesives) in 6 test runs per each seasoning variant (A-F variants, Table 5-10). The total amount of tested epoxy adhesive samples amounted to 240 items”).
Section 3.1: in this section the author only discussed what samples presented the better behaviour without discuss which caused this behaviour. A deeply discussion about the effect of temperature and time on the chemical structure of the samples is necessary. In addition, the author must compare the results of this work with others from the literature.
Response: Some information was added or/and changed.
Figure 4, Figure 5 and Figure 6: the author must discuss the effect of using CaCO3 and the effect of the different curing agents on the properties evaluated.
Response: Some information was added or/and changed.
Section 3.2: the discussions in this section must be improved. What the usage of different curing agents caused in the properties evaluated? What the effect of the different chemical structure of the curing agents on the properties? What is the effect of using CaCO3? Moreover, the effect of temperature and time on the ageing of the samples?
Response: Some information was added or/and changed.
Conclusions section: please improve this section based on the above comments. In addition, remove lines 489 to 492.
Response: I try to improve the conclusions section and lines 489 to 492 were removed.

Reviewer 2 Report
The paper titled "the impact of seasoning conditions on mechanical properties of modified and unmodified epoxy adhesive compounds" deals with testing and epoxy resin, Epidian 53, cured with two different curing agents, IDA and Z1 and adding calcium cabonate.
Even if the topic can be of interest, the paper has serious flaws and I cannot recommend publication because it seems more a technical report than a scientific paper
Major concerns are
1) English language must be completely checked and revised, too many mistakes are present
2) In a scientific paper, especially if for publication in Polymers, the nature of polymers and other reagents must be well defined. The resin is a commercial Epidian 52 and the author only states that it is an epoxy resin styrene-modified with no further characterization. Furthermore the chemical nature of the curing agents is completely unknown. Readers cannot understand the chemical rationale under the study proposed.
3) Author presents two SEM photos (Figure 2) and correctly states that the non homogeneity of the resin is evident due to filler agglomeration. But nothing more is added: which is the consequence? why no further study to lower agglomeration were conducted?
4) Table 4 and the following text are in the form of a technical report more than in the form of a scientific paper.
5) The first time an acronym is used, the full name of the chemical must be given: in lines 51, 55-56, just to make some examples, acronyms are given with no indication at all about the name of the compounds
Author Response
Response – Reviewer 2
Thank you for all comments.
I try to consider all Reviewer comments.
Some information was change and deleted.
All changes are in blue.
I would like to answer to the following questions.
1) English language must be completely checked and revised, too many mistakes are present
Response: I try to improve English language.
2) In a scientific paper, especially if for publication in Polymers, the nature of polymers and other reagents must be well defined. The resin is a commercial Epidian 52 and the author only states that it is an epoxy resin styrene-modified with no further characterization. Furthermore the chemical nature of the curing agents is completely unknown. Readers cannot understand the chemical rationale under the study proposed.
Response: In the manuscript selected physicochemical properties of Epidian 53 epoxy resin (not Epidian 52) was presented in the Table 2 and I think that some general information was presented in the text. The Epidian 53 is epoxy resin - reaction product: bisphenol-A and epichlorhydrin.
Some information was added in the text and some detailed information was presented in the BN-73/6376-01 branch standard.
And below is characteristic this epoxy resin.
This liquid epoxy compound has colours ranging from yellow to dark-brown and a distinctive odour of aromatic hydrocarbons. Epidian 53 is produced by thinning epoxy resin Epidian 5 with styrene. It exhibits a low viscosity and average reactivity. The application of a high temperature during the curing process for this resin accelerates polyreaction. The curing process for this resin takes place after the addition of a suitable amount of a curing agent and consists of three stages (when curing at room temperature). The first stage is an initial cure period and it takes 6 – 8 hours approx.; then, after 72 hours, the degree of curing is approx. 80 – 90 %. The total cure time is 7-14 days. The application of a high temperature in the first stage of the curing process results in accelerated polyreaction. As a result of curing, this resin becomes a good electro-insulating material. When used as an adhesive, this resin has the highest shear strength provided that the curing process is performed at approx. 110oC (even up to 22.5 MPa).
3) Author presents two SEM photos (Figure 2) and correctly states that the non homogeneity of the resin is evident due to filler agglomeration. But nothing more is added: which is the consequence? why no further study to lower agglomeration were conducted?
Response:
The presented SEM images were intended only for the general characterization of the adhesive compositions obtained. It is noted that the non-homogeneity of the resin is evident due to filler agglomeration. Further analysis of the method of preparing the adhesive composition, including the mixing method, will be presented in further publications in which attention will be focused on the methodology of preparing adhesive compositions modified with calcium carbonate. For this reason, this point has been included in the research methodology, as a characteristic of adhesive compositions modified with calcium carbonate, and not as test results. Although, as suggested by the second reviewer, this information should be transferred to the point – results and discussion section (“Figure 2: this figure and the discussion about this figure are results and must be added to the results and discussion section.”).
Due to the fact that this is not the subject of the analysis in this article, I decided that these messages will remain in item 2, as an additional description of the analyzed compositions, allowing to display this composition in the hardened state.
4) Table 4 and the following text are in the form of a technical report more than in the form of a scientific paper.
Response:
The information presented in Table 4 could also be presented in the form of a text. It seemed to me that it was more readable. From the technological point of view, it is clear information presenting both the individual operations of sample preparation of adhesive compositions as well as their detailed description along with technological parameters. If the Reviewer thinks that the information in Table 4 should be presented in the form of a description, I can of course change it, although in my opinion this form is much more legible. Therefore, I am asking you for information as to whether the reviewer accepts my explanation or does he think that the information contained in table 4 should be presented in the form of a text.
5) The first time an acronym is used, the full name of the chemical must be given: in lines 51, 55-56, just to make some examples, acronyms are given with no indication at all about the name of the compounds
Response: Acronym “AV119” is the adhesive symbol/designation adopted by the adhesive manufacturer and it is difficult to provide the full name of the chemical. The information was presented on the basis on Ref. [25]. The description of the adhesive e.g. “single-part structural epoxy past adhesive AV119” is presented in the text.
The same source (from references - Ref. 17) concerns the acronyms ODCB and CDCB. In this case the full name of type of adhesive joins was added, eg.: open-faced double cantilever beam (ODCB) specimens and degraded closed double cantilever beam (CDCB) specimens.

Round 2
Reviewer 1 Report
After corrections the manuscript is suitable for publication in Polymers journal.
Reviewer 2 Report
The paper has been significantly improved, changing most of the points I criticized. Only some minor english mistakes are now present
Nevertheless, given the subject of the paper and the very general chemical characterization of the resin and curing agents, probably a journal such as "Materials" would be more appropriate